# Low-Dose Radiation Therapy (LDRT) against Cancer and Inflammatory or Degenerative Diseases: Three Parallel Stories with a Common Molecular Mechanism Involving the Nucleoshuttling of the ATM Protein?

**DOI:** 10.3390/cancers15051482

**Published:** 2023-02-26

**Authors:** Eymeric Le Reun, Nicolas Foray

**Affiliations:** Inserm, U1296 Unit, “Radiation: Defense, Health and Environment”, Centre Léon-Bérard, 28 rue Laennec, 69008 Lyon, France

**Keywords:** low-dose, radiation therapy, LDRT, radiosensitivity, cancer, inflammation, Alzheimer’s disease, HRS, hormesis, adaptive response, ATM protein

## Abstract

**Simple Summary:**

We reviewed the medical applications of low-dose radiation therapy (LDRT) in cancer and inflammatory and degenerative diseases by proposing a unified mechanistic model based on the radiation-induced nucleoshuttling of the ATM kinase to better understand the interest in low-dose of radiation.

**Abstract:**

Very early after their discovery, X-rays were used in multiple medical applications, such as treatments against cancer, inflammation and pain. Because of technological constraints, such applications involved X-ray doses lower than 1 Gy per session. Progressively, notably in oncology, the dose per session increased. However, the approach of delivering less than 1 Gy per session, now called low-dose radiation therapy (LDRT), was preserved and is still applied in very specific cases. More recently, LDRT has also been applied in some trials to protect against lung inflammation after COVID-19 infection or to treat degenerative syndromes such as Alzheimer’s disease. LDRT illustrates well the discontinuity of the dose-response curve and the counterintuitive observation that a low dose may produce a biological effect higher than a certain higher dose. Even if further investigations are needed to document and optimize LDRT, the apparent paradox of some radiobiological effects specific to low dose may be explained by the same mechanistic model based on the radiation-induced nucleoshuttling of the ATM kinase, a protein involved in various stress response pathways.

## 1. Introduction

### 1.1. Historical Features of Low-Dose Radiotherapy

Several months after the discovery of X-rays, radiation pioneers performed the first clinical attempts of radiation therapy (RT) against a plethora of diseases, including tuberculosis [1], cancer [2,3,4], psoriasis, scleroderma, abscesses, and inflammatory and degenerative disorders [5,6,7]. However, from 1896 to the 1930s, the medical applications of X-rays were necessarily limited by technological constraints. For example, the notion of dose was not defined yet. The X-ray tubes were fragile and did not deliver high doses of high-energy X-rays but rather low doses (<1 Gy) of soft X-rays (<100 kV). Hence, the first successes in RT were attributed either to spectacular but isolated effects after a single exposure to a low dose (and therefore needed to be more documented) or to hyper-fractionated RT of low doses (raising questions about the biological effects of the fractionation of the dose, on the one hand, and, on the other hand, the existence of radiobiological phenomena specific to low doses). As an example, the reconstitution of the first documented RT against cancer performed by Victor Despeignes in July 1896 revealed that the maximal dose delivered to the tumor per session was lower than 0.8 Gy, while the volume of the tumor decreased significantly [8,9]. Today, we know that the efficiency of this historical treatment could only be obtained on a very radiosensitive gastric lymphoma eliciting hypersensitivity to low-dose phenomenon [9]. Some years after, Claudius Regaud succeeded in sterilizing ram testes without any acute skin reaction with a repeated low dose of soft X-rays, ensuring the balance between a significant biological effect of low (sublethal) doses without the deleterious consequences of a too-high cumulative dose [10,11]. The notion of sublethal dose led to a large number of applications of X-rays to cure diseases.

After the 1930s, RT based either on a total dose or a dose per session lower than that applied in standard RT to date was progressively applied in the management of numerous cancers and non-malignant diseases [4,7]. However, the molecular and cellular mechanisms supporting the capacity of sublethal doses to cure such pathologies remain to be elucidated, inasmuch as the proposed biological interpretations were indifferently based on immunologic or inflammatory features and were not convincing [4,7].

From the 1940s, the investigations into the Hiroshima survivors documented some radiobiological phenomena specific to low doses and generated controversies about the non-linear threshold (NLT) and the linear non-threshold (LNT) models, suggesting that, in the first case, a range of a low dose should exist in which the risk of radiation-induced cancer is either negligible or not measurable and, in the second case, the risk of cancer is directly proportional to the dose [12]. Particularly, in this period, the hormesis phenomenon describing the beneficial effect of very low doses (lower than 25 mGy) was reviewed, and the role of this phenomenon in radiation protection was debated [13,14]. However, again, the radiobiological interpretation of this phenomenon remained to be elucidated.

In the next decades, notably in the 1980s, two other radiobiological phenomena specific to low doses were pointed out: the adaptive response (AR), consisting of a protective effect caused by a “priming” low dose (typically ranging from 1 mGy to 0.5 Gy) delivered after a period of time (ranging from 2 to 24 h) before a higher “challenging” dose (typically of the order of some Gy) [15,16], and hyper-radiosensitivity to low doses (HRS), consisting of a deleterious effect generally produced between 0.1 to 0.8 Gy at a high dose rate (>1 Gy/min) but equivalent to the effect provided by a dose 5 to 10 times higher [17]. Again, the radiobiological interpretation of these phenomena remained to be elucidated [16,18].

### 1.2. The Molecular Features of Hormesis, AR, and HRS Phenomena Explained by A Unified Model

The hormesis, AR, and HRS phenomena are likely to be involved in the molecular and cellular response to RT based on the deliverance of several fractions of Gy per session. Hence, a better knowledge of these specific radiobiological phenomena is essential for our understanding of the medical interest of such a specific type of RT. Recently, in the frame of the model based on the radio-induced nucleoshuttling of the ATM protein (RIANS), a unified interpretation of the hormesis, AR, and HRS phenomena has been proposed [16,19,20] (Figure 1). Briefly, in quiescent normal and cancer cells, ATM homodimers are present in the cytoplasm. Exposure to radiation produces, in a linearly dose-dependent manner, oxidative stress leading to DNA double-strand breaks (DSB) [21] and the monomerization of cytoplasmic ATM homodimers [20]. The ATM monomers then diffuse into the nucleus and mediate DSB recognition and repair via non-homologous end-joining (NHEJ), the most predominant DSB pathway in mammalians [22,23]. Any delay in the RIANS leads to unrecognized DSB, which may contribute to the lethal cellular effect (*radiosensitivity*) via non-repaired DSB, carcinogenesis (*radiosusceptibility*) via misrepaired DSB and/or accelerated aging (*radiodegeneration*) via the accumulation of tolerated DSB [23,24,25]. A delayed RIANS is generally caused by the formation of cytoplasmic multiprotein complexes between ATM and ATM substrates called X-proteins. It has been shown that the proteins phosphorylated by the serine/threonine ATM kinase elicit some SQ or TQ domains [26]. Hence, any cytoplasmic protein holding SQ or TQ can be considered as a potential X-protein [23,24,27]. Some heterozygous mutations of X-proteins lead to their overexpression in the cytoplasm. From the RIANS data, three groups of individual response to high doses of ionizing radiation have been defined [23]: group I gathers radioresistant cells with a fast RIANS (some minutes after 2 Gy) (Figure 1a); group II gathers radiosensitive cells with a delayed RIANS (some hours after 2 Gy) (Figure 1b); and group III gathers hyper-radiosensitive cells either with lack of RIANS such as the *ATM*-mutated cells or with a normal RIANS but a gross DSB repair such as the *LIG4*-mutated cells (Figure 1c). From the bases of the RIANS model, the following interpretations of the hormesis, AR, and HRS phenomena have been proposed:*Hormesis*: One Gy X-rays generally induce about 40 DSB and 10^5^ to 10^6^ ATM monomers per human non-transformed cell. Consequently, very few DSB are induced per cell at doses lower than 1/40 Gy (i.e., 25 mGy). If some spontaneous DSB are present in cells, exposure to a dose lower than 25 mGy provides some ATM monomers that may contribute to recognizing and repairing these spontaneous DSB. Hence, if these spontaneous DSB represent a risk of cancer or aging, a dose lower than 25 mGy may decrease such a risk. This phenomenon is generally observed in group I cells since the amount of ATM monomers after irradiation is significant at such low doses by comparison with group II cells, in which the flux of ATM monomers is reduced, and in group III cells, in which DSB recognition or repair are impaired [19] (Figure 1d).*Adaptive response*: In the AR scenario, a “priming” (low) dose may produce few DSB but overall a significant amount of ATM monomers that may contribute to recognizing and repairing the DSB induced by a “challenging” (high) dose, as far as the time interval between the two doses preserves the activity of the ATM monomers in the nucleus. The AR phenomenon is generally observed in group II cells since the contribution of ATM monomers provided by the priming dose is too low (due to their sequestration in the cytoplasm by the X-proteins) to recognize all the DSB induced by the priming dose. Conversely, in group I cells, all the DSB are recognized by the high flux of ATM monomers that diffuse in the nucleus [19].*Hypersensitivity to low dose (HRS)*: As said above, for the priming dose, a low dose induces few DSB and few ATM monomers. In group II cells, the sequestration of ATM monomers by overexpressed X-proteins may drastically reduce the number of ATM monomers that finally diffuse in the nucleus. Consequently, in group II cells, after a single low dose (whether after the priming dose in the frame of AR scenario or after an HRS dose), the few DSB induced by the low dose may not all be recognized and repaired by the few ATM monomers available in the nucleus. The rate of unrepaired DSB after a low dose (e.g., 0.2 Gy) could produce an effect equivalent to that produced by a dose 5 to 10 times higher (e.g., 2 Gy) [20]. Again, in group I cells, such conditions are never reached since there are no ATM–X-protein complexes, and the amount of ATM monomers is always sufficient [20,23] (Figure 1e).

Throughout the history of RT, some authors have proposed to apply RT based on a total dose lower than the standard (the most current) treatment plans. Such an approach that consists of reducing the dose per session or the number of sessions itself may be described under the general term of low-dose radiation therapy (LDRT). Considering, on the one hand, the increasing importance of LDRT in the treatment of some cancers and inflammatory and degenerative diseases and, on the other hand, the emerging interest in the RIANS model that may provide a unified mechanistic interpretation of the effects caused by the specific low-dose phenomena, this review aims to establish:A state of the art of the application of LDRT in clinical practice;A review of the basic mechanisms supporting the application of LDRT;A unified RIANS model integrating the radiobiological bases of LDRT.

## 2. LDRT in Oncology

### 2.1. Clinical Data about LDRT against Cancer

As evoked above, the technological limits of the 1895–1930 period did not facilitate the application of high doses and high-energy photons. Hence, the first attempts of anti-cancer RT were performed with single and repeated low doses [2,3,8,9]. LDRT has long been applied for the treatment of acute leukemia, with myeloablative total body irradiation (TBI) prior to hematopoietic stem cell transplantation. A TBI of 12 Gy total is commonly delivered according to different schedules: 2 Gy per session twice daily for 3 days, 3 Gy per session daily for 4 days, and, considering low doses, 1.5 Gy per session twice daily, or 1.2 Gy per session three times daily for 4 days [28].

Based on the good results of TBI in non-solid tumors, and since lymphomas are known to be radiosensitive [29], LDRT has then been assessed in patients with non-Hodgkin’s lymphoma (NHL). In a 1970s cohort, the TBI of 68 patients with NHL, with a median dose of 0.1 Gy per session and mean total dose of 1.78 Gy, achieved a recurrence-free survival of 32% and 27%, respectively, at 5 and 10 years [30]. A decade later, Richaud et al. underlined the excellent remission rate in NHL patients who received two courses of LD-TBI (0.75 Gy per 5 sessions) with concomitant chemotherapy followed 1 month later by radical involved-field RT [31].

More recently, LDRT exhibited a clinical interest in treating metastatic diseases. In 2004, a study examining 36 metastatic tumor nodules after a 12-day RT revealed that time to regrowth was longer when using an ultra-fractionated scheme (0.5 Gy every 4 h) compared to a standard fractionation (1.5 Gy daily) [32]. In a phase II study, Morganti et al. demonstrated that 38.9% of patients with metastatic colorectal cancer showed a complete response in lesions treated with two daily LDRT sessions (20 cGy every 6 h) on each of the 12 concomitant chemotherapy cycles [33]. Combined with immunotherapy, LDRT of 1 Gy per session also resulted in metastasis (liver and inguinal lymph node) complete response in an elderly patient with vaginal melanoma [34]. In another phase II trial, 74 patients with metastatic cancer (non-small cell lung cancer, *n* = 38; melanoma, *n* = 21) underwent a treatment of high-dose RT (HDRT: 20–70 Gy total; 3–12.5 Gy per session) associated or not with LDRT (1–10 Gy total; 0.5–2 Gy per session). Follow-up showed that adding LDRT to HDRT strengthened the 4-month disease control response (47% in HDRT + LDRT versus 37% in HDRT alone; *p* = 0.38) and increased the overall response (26% in HDRT + LDRT versus 13% in HDRT alone; *p* = 0.27). This scheme also appeared to be safe, with only three patients (4.1%) suffering from toxicity of grade 3 or more. However, it must be mentioned that the patients enrolled in the study received immunotherapy [35]. It is noteworthy that the hyperfractionated schemes commonly used in radiation oncology concern doses from 1.1 Gy per session (head and neck squamous cell carcinomas, [36]) to 1.5 Gy per session (limited small-cell lung cancer [37]), which is, strictly speaking, actually not in the historical scope of LDRT.

Table 1 summarizes the clinical attempts and results of LDRT in oncology. However, it must be stressed that the available data are mostly based on case reports or trials with a relatively low number of patients.

### 2.2. Biological Hypotheses about the Anti-Cancer Effect of LDRT

The application of LD-TBI against lymphoma was the result of a long succession of empirical attempts that was facilitated by the current radiosensitivity of lymphoma. In a second step, from the 1980s, the HRS phenomenon became the major molecular basis of LDRT against lymphoma. In fact, HRS has been described in a large subset of tumor cell lines. Particularly, Joiner and Marples observed an increased radiosensitivity in murine cells below 1 Gy per session, which was not predicted by the standard linear-quadratic (LQ) model that is still the current mathematical description of the cellular radiation response [20,38,39,40]. In fact, the standard LQ model predicts that clonogenic cell survival progressively and continuously decreases as far as the radiation dose increases. Interestingly, survival curves showing the HRS phenomenon elicited a discontinuous dose-response suggesting an excess of cell death in a specific range of low doses [20,40]. Later, Lambin et al. described excess mortality in human tumor cells after X-irradiation between 0.05–1 Gy, both in colorectal cancer [17] and bladder carcinoma [41], and, once again, underestimated by the LQ model. In parallel, one human glioblastoma cell line was found significantly more sensitive to LDRT (3 sessions of 0.4 Gy) than with a unique session of 1.2 Gy [42]. Observations showed that, in HRS-positive cells irradiated in the G2 phase, DSB were normally recognized by ATM phosphorylation of the histone variant H2AX (γH2AX) [43], but cell cycle arrest was ineffective [44], leading to excess tumor cell death.

In addition to HRS, the immune microenvironment has also been mentioned to explain the LDRT anti-cancer efficiency. Some studies described an increased infiltration of effector immune cells (T cells and NK cells) in animal [45,46] and human [33,47] tumors submitted to LDRT. However, such an explanation has not been documented in a large subset of tumors, and the causal link between the immune microenvironment and the HRS remains to be established.

### 2.3. The HRS Phenomenon and the RIANS Model 

As evoked in the Introduction, the number of radiation-induced DSB and ATM monomers obey a linearly dose-dependent law, whatever the group of radiosensitivity of cells [20,23]. In the group I (radioresistant) cells, there are always many more ATM monomers than DSB produced, which facilitates the recognition of all the radiation-induced DSB by the ATM monomers available in the nucleus after irradiation [20,23]. In group III (hyper-radiosensitive) cells, the HRS phenomenon does not exist since there are no active ATM monomers to recognize DSB or no active NHEJ pathway to repair them. Conversely, in group II cells, the recognition of DSB is incomplete since the overexpressed X-proteins sequestrate a large amount of ATM monomers in the cytoplasm [20]. In these conditions, the subset of non-recognized DSB may produce a significant tumor cell-killing effect. The optimal conditions of LDRT may be simulated in vitro. At a high dose-rate (> 1 Gy/min), the maximal HRS effect may represent a decrease in cell survival of 25% in the range of 0.1 to 0.8 Gy [48]. Hence, an LDRT consisting of the application of several doses belonging to this dose range may result in a significant tumor cell-killing effect. However, the benefit of the LDRT is conditioned by the fact that the tumor should either belong to group III (hyper-radiosensitive) or be HRS-positive and belonging to group II and that the surrounding healthy tissues should be HRS-negative and group I (radioresistant) to avoid any adverse reactions or risk of radiation-induced cancer. It is also noteworthy that the dose range in which HRS is maximal decreases with the dose-rate [48]. Hence, further investigations are needed to optimize the LDRT protocol (doses and dose-rate) to reach the maximal tumor cell-killing effect by sparing healthy tissues.

## 3. LDRT in Inflammation-Related Pathologies

### 3.1. Clinical Data about LDRT against Inflammation in Rheumatology

The palliative properties of X-rays have been abundantly documented together with their anti-cancer efficiency: indeed, reducing tumor volume can be frequently accompanied by the reduction of pain due to the decompression of the tumor against surrounding tissues. During the first historical attempt at RT, about 6 months after the discovery of X-rays, Victor Despeignes described in his X-ray-treated patient a significant reduction of pain together with the reduction of the tumor [2,3,4]. However, the analgesic properties of X-rays have been early described on other diseases than cancer, notably on rheumatology diseases and particularly arthritis. Indeed, radiation pioneers such as Sokolow (1897), Stenbeck (1899), Albers-Schönberg (1900), Stembo (1900), and Moser (1903) described a significant reduction in pain following RT against acute arthritis [5,6,7,49,50]. As already evoked in the Introduction, in the first decade of the 20th century, X-ray treatments against arthritis resulted in single or fractionated X-ray sessions of low doses because of technology limitations. After the 1930s, a number of anti-inflammatory RT protocols flourished, and the total dose increased. In parallel with the radium girls [51], the risk of radiation-induced cancers and/or long-term sequelae suggested that the total dose should be limited [52]. It is noteworthy that RT against ankylosing spondylitis became a routine from the 1930s up to the 1950s [53].

To date, the recommended total dose for painful joint disorders, such as synovitis, elbow syndrome, shoulder syndrome, trochanteric bursitis, plantar fasciitis, and arthrosis (although osteoarthritis is not always associated with inflammation), is typically set around 3–6 Gy, delivered in 0.5–1 Gy per session [54]. In a prospective study, Micke et al. depicted a pain relief effect of LDRT (6 Gy delivered in 0.5–1 Gy per session) in 166 elderly patients with degenerative joint disorders [55]. A more recent study came to the same conclusion after irradiating 196 patients with painful osteoarthritis of the ankle and foot with a LDRT scheme (3–6 Gy delivered in 0.5–1 Gy per session) [56]. Conversely, three randomized sham-controlled clinical trials invalidated this supposed LDRT benefit in patients with knee and hand osteoarthritis [57,58,59]. Altogether, this short review suggests that LDRT against inflammatory bone and joint tissue diseases was an important application of X-rays very early after the discovery of X-rays. However, the mechanisms by which LDRT can act on the bone system are still difficult to determine since the response of the regulation between bone formation by osteoblasts and bone resorption by osteoclasts appears to be strongly dose-dependent. Low doses were shown to modulate the immune system and impact bone homeostasis. Furthermore, low doses may stimulate osteoblast proliferation and/or differentiation while they inhibit the viability of osteoclasts, which may facilitate the promotion of bone formation and the reduction of bone resorption. Hence, further investigations are still needed to build a unified model linking the immunological and molecular features of the bone response to radiation [60].

### 3.2. Clinical Data about LDRT against Inflammation after COVID-19 Infection

Recently, another important application of LDRT emerged with the coronavirus disease 2019 (COVID-19) pandemic. COVID-19, due to the severe acute respiratory syndrome coronavirus 2 (SARS-CoV-2), is responsible for about 6.8 million deaths worldwide as of 3 February 2023 according to the WHO COVID-19 dashboard (https://covid19.who.int/, accessed on 1 February 2020). Lung inflammation has been mentioned to play a major role in COVID-19 pathogenesis and its mortality [61,62,63].

Since the beginning of the coronavirus pandemic, many authors evaluated the use of LDRT in COVID-19 patients, probably inspired by the anti-inflammatory properties of X-rays from the above data and/or by empirical observations after CT scan exams [62,64]. In two prospective trials, a single dose of 0.5–1 Gy to whole lungs produced a significant clinical improvement in terms of pulse oximetric saturation (SatO_2_)/fraction of inspired oxygen (FiO_2_) ratio without radiotoxicity [65,66]. In a phase II trial, Ganesan et al. reported some pulmonary benefits (the SatO_2_/FiO_2_ ratio significantly improved, and the oxygen supply decreased significantly) in 88% of COVID-19 patients after LDRT delivered in a single session of 0.5 Gy to the bilateral whole lungs [67]. These results were in agreement with those published by Rodríguez-Tomàs et al. in 30 patients submitted to the same single 0.5 Gy LDRT scheme [68], but not with a randomized trial delivering 1 Gy in 20 patients [69]. Two randomized ongoing clinical trials are currently evaluating other dose fractionations: 1.5 Gy versus sham (NCT04433949, Emory University, USA, with its interim analysis [70]) and 0.35 Gy or 1 Gy versus sham (NCT04466683, Ohio State University Comprehensive Cancer Center, USA). Altogether, the clinical attempts and results of LDRT in inflammation-related pathologies (rheumatology and COVID-19) are presented in Table 2.

In a recent overview, by insisting again on the low amount of data available (number of treated patients, variety of the treatments, and the diversity of the molecular and cellular endpoints used), Rödel et al. (2020) stressed the importance of the appropriate timing of irradiation with the dose fractionation and the succession of the different steps observed during a COVID-19 infection. Hence, even if LDRT appears to be efficient against inflammation after COVID-19 infection, strict monitoring and disease phase-adapted treatment based on objective markers such as IL-6 and D-dimer in serum are required [71].

### 3.3. Biological Hypotheses about the Anti-Inflammatory Effect of LDRT

It is noteworthy that, although inflammation may follow some steps common to both arthritis and COVID-19, a molecular model may not necessarily be relevant for both pathologies. However, hyper-inflammation has been documented both in rheumatology [72,73] and in COVID-19 [74]. On this point, cells involved in the inflammation process could represent a target for LDRT. In serums of osteoarthritis mice models, Weissmann et al. observed a significant reduction in the pro-inflammatory CD8+ T-cells and IL-17A and a significant increase in the anti-inflammatory CD4+ T-cells, IL-4, and IL-6 after LDRT (0.5–1 Gy per session) [56]. Rödel et al. and Arenas et al. showed that LDRT (investigated dose range 0.3–0.7 Gy) significantly reduces the adhesion of peripheral blood mononuclear cells and the number of adherent leukocytes, both in vitro and in vivo [75,76,77]. Furthermore, some authors observed a more pronounced production of anti-inflammatory TGF-β1 in blood [75] and a reduction in lymphocyte counts [69] after LDRT. Interestingly, Sanmamed et al. described a significant decrease in inflammation biomarkers such as C-reactive protein and lymphocyte counts after 1 Gy in the lungs [66]. In the same way, a single dose of 0.5 Gy applied to the lungs induced lower C-reactive protein and TGF-β1 concentrations in COVID-19 patients [68]. X-irradiation between 0.5–1 Gy also exhibited a significant reduction of CCL20, a chemokine that facilitates the recruitment of polymorphonuclear neutrophils toward the inflammation sites and is dependent on TGF-β1 [78]. More recently, in murine models, Meziani et al. (2021) showed that LDRT increases the immunosuppressive profile of lung macrophages during viral infection and pneumonia [79]. Jackson et al. showed that LDRT against acute inflammatory lung injury suppresses the bleomycin-induced accumulation of pulmonary interstitial macrophages [80].

Altogether, this brief review suggests that, for both treatments against arthritis and COVID-19, LDRT results in a reduction in pro-inflammatory pathways and the stimulation of anti-inflammatory pathways, even if the specificities and the optimization of each type of treatment remain to be determined.

### 3.4. Tissue Inflammation and the RIANS Model

Inflammation involves a myriad of proteins through a complex process of autocrine, paracrine, and endocrine secretions of cytokines that may serve as pro- or anti-inflammatory actors. Notably, IL-1b, IL-6, and TNF-α are rather pro-inflammatory cytokines, while the IL-1 receptor antagonists IL-13 and TGF-β are rather anti-inflammatory cytokines [81]. Interestingly, all these cytokines are expressed in a radiation dose-dependent manner. Furthermore, these cytokines are cytoplasmic, overexpressed after exposure to radiation, and hold some SQ/TQ domains, preferentially phosphorylated by the ATM kinase [26]. Altogether, these statements suggest that ATM is strongly involved in the inflammatory pathways via interactions and phosphorylations of these cytokines that may be therefore considered as potential X-proteins. Particularly, activation of IL-6 was shown to promote the proliferation of glioblastoma cells after ionizing radiation in the presence of functional ATM [82], while activation of TGF-β was shown to consolidate the ATM-CHK2-p53 complex, favoring a progressive senescence death rather than a proliferation of cells that would increase in the number of propagated DNA damage and therefore the toxicity rate [83]. These two representative examples showed that the ATM kinase is strongly involved in the fate of irradiated cells via interactions with (and phosphorylations of?) pro-inflammatory cytokines, which stimulates cell proliferation and therefore amplification and propagation of DNA damage, mitotic death, and apoptosis. Conversely, via interactions with (and phosphorylations of?) anti-inflammatory cytokines, ATM promotes the cell cycle checkpoint arrests and senescence death, which limits the amount of unrepaired DNA damage. Unfortunately, there are still no quantitative evaluations of the expression of cytokines to be compared with the number of available ATM monomers in a dose range corresponding to LDRT. However, despite of this lack of data, the RIANS model may predict the influence of individual factors in the response to low doses since individual X-proteins and cytokines as radiation-induced X-proteins may contribute together to the sequestration of ATM in the cytoplasm.

Further investigations are therefore needed to better clarify the role of ATM in the inflammatory process. However, it is noteworthy that if the HRS phenomenon occurs, it also affects the expression of cytokines, therefore complexifying the dose-dependence of the inflammatory process.

Lastly, it is important to note that viral infection is generally associated with the expression of viral proteins. In the particular case of COVID-19, the Spike protein may hold between 5 to 13 SQ/TQ domains according to the nature of the associated coronavirus. Overexpressed in cytoplasm during viral infection, the Spike protein appears to be a potential X-protein that may sequestrate ATM and drastically reduce the diffusion of ATM monomers in the nucleus to respond to DNA strand breaks induced by the viral infection [84]. Hence, the individuals of group II, with a delayed RIANS, may be at risk of toxicity during infection. Again, further investigations are needed to document the role of ATM during the viral infection process, but it is likely that the RIANS model may contribute to understanding the basic mechanisms and to predicting the cases at risk.

## 4. LDRT in Alzheimer’s Disease (AD)

Alzheimer’s disease (AD) is the world’s leading cause of dementia, considered by the World Health Organization (WHO) as a global public health priority. Most of the time, only a probability diagnosis can be made on the basis of clinical arguments (age, hippocampal amnesia, and family history of AD) [85], supported by biological criteria after lumbar puncture (increased pTau protein and decreased Aβ1-42 peptide in the cerebrospinal fluid) [86] and cerebral imaging (cortico-subcortical atrophy, hippocampal atrophy, and no evidence of other cause of dementia) [87]. Definitive diagnosis of AD is based on the histological demonstration of extracellular amyloid-Aβ42 peptide deposits (senile plaques) in the supporting tissue of the central nervous system [88] and intracellular neurofibrillary tangles of hyperphosphorylated Tau protein [89]. If neurofibrillary degeneration and senile plaques can be found in other pathologies, their association is pathognomonic of AD. Neuroinflammation is also a frequent sign encountered in the pathogenesis of AD [90,91,92].

### 4.1. LDRT Clinical Trials in AD Patients

Since 2016, Cuttler et al. published two case reports describing patients with AD whose cognitive performance temporarily improved after iterative low-dose brain computed tomography (CT) scans, about 40–50 mGy each within 10 s [93,94,95]. The first case report presents an 81 years-old woman with AD progressing for 10 years, living in hospice care for several months, «almost totally noncommunicative», «completely nonresponsive», «frequently refusing her medications», «almost immobile», and who did «not attempt to rise from her wheelchair in months», when examined in May 2015 by a neuropsychologist. In July, her physician prescribed two CT scans of the brain (total dose of 82 mGy) to exclude any anatomical changes and, according to the patient’s spouse, to «stimulate neuroprotective systems». The patient improved so dramatically (cognition, speech, movement, and appetite) in the following days that she underwent three other CT scans in a 3-month period: a third (39 mGy) 2 weeks after the previous, a fourth (47 mGy) 2 weeks later, and a fifth (39 mGy) 6 weeks later in October. Consecutively, her condition improved enough that she no longer required to stay in hospice care, and she moved to a home for seniors on November 2015 with a stimulating day program. The patient had a sixth brain CT scan (40 mGy) 5 months after the fifth, a seventh (40 mGy) 4 months later, and an eighth (40 mGy) 4 months later, and as soon as her state began declining, she had additional brain CT scans earlier, with a ninth (40 mGy) 47 days later, and the last two—the tenth and eleventh—(80 mGy) 41 days later, in January 2017. Ultimately, she received about 0.45 Gy over thw whole brain in 11 CT scans in an 18-month period. She finally died on 18 May 2018, 16 months after the last CT [88,89].

The same authors reported another case of a 73-year-old man diagnosed with AD in January 2015, who benefited from similar brain CT scans but associated with the Mini-Mental State Examination (MMSE) throughout [95]. It is noteworthy that an MMSE score of 23/30 or less generally signs the presence of cognitive impairment [96]. The patient presented an MMSE baseline score of 22/30 in February 2016 and had a primary CT scan of 46 mGy in April 2016. Three weeks after, his MMSE improved to 24/30. He received a second CT scan two months after the first, and his MMSE continued to increase to 26/30 when assessed in September. The third and fourth CT scans preceded another 26/30 MMSE score in February 2017, which then started to decrease down to 25/30 in March and 24/30 in April. Two additional CT scans (a fifth and sixth) were given until October 2017, when the patient had been considered to enter the moderate or mid-range of AD. He then progressively declined and finally died on 7 September 2020, at age 77.

Radiation effects on the brain are actively evaluated in vivo. Hence, X-irradiation (5 × 2 Gy) of the brain seems to be correlated with cognitive improvement in murine AD models [97,98,99]. Based on these encouraging LDRT results on animals, some clinical trials are conducted with such dose schemes. Hence, NCT02359864 (William Beaumont Hospitals, Royal Oak, MI, USA) and NCT02769000 (Virgina Commonwealth University, Richmond, VA, USA) attempted 5 × 2 Gy and 10 × 2 Gy fractionations but were suspended for staffing and budget limitations and interrupted due to COVID-19, respectively. Finally, the randomized and prospective NCT03352258 (Geneva University Hospital, Geneva, Switzerland) is currently recruiting, with the aim to treat 10 AD patients at 5 × 2 Gy. However, since human cells are more radiosensitive than murine cells [100,101], it is possible that the dose employed in these trials could be reduced to achieve clinical efficiency. It appears urgent to better explain and justify the drastic differences observed between the studies implying CT scan and mGy doses and those implying fractions of 2 Gy with radiotherapy irradiators. Again, the schedule of the dose fractionation should be adapted to the kinetics of the disease progression.

In continuity with their two questioning case reports [93,95], Cuttler et al. undertook a single-arm study in four AD patients with severe but stable dementia (age: 81–90 yr, MMSE baseline score: 0, 0, 0, and 5, respectively) [102]. Patients received two brain CT scans at once during the same treatment session (80 mGy total), then a third (40 mGy) two weeks later, and a fourth and last (40 mGy) two more weeks later. Three of the four patients showed slight cognition and behavior improvements on quantitative measures; notably, a Severe Impairment Battery (SIB) score improved from 21% to 31% in one patient and basic activities of daily living (ADL) improved from 26.7% to 30% in another patient. Although these minor results produced no long-term benefit for patients, some of the brief improvements observed directly after CT scans could reflect a significant biological effect of LDRT. Clinical attempts and results of LDRT in AD are depicted in Table 3. Lastly, if cellular mechanisms of LDRT are emerging in AD models, there is still no molecular model to explain LDRT effects in human AD patients.

### 4.2. Biological Hypotheses about the Beneficial Effect of LDRT for AD

In two in vivo murine studies, cognitive alleviation after LDRT was also associated with a reduction in amyloid-Aβ42 plaques [98,99]. Furthermore, Yang et al. pointed out that X-rays positive effects on AD (senile plaque control and cognitive improvements) could be reached even when decreasing the dose in mice from standard fractionation (5 × 2 Gy) to an LDRT scheme (5 × 0.6 Gy) [99]. The radio-induced reduction of amyloid-Aβ42 plaques in AD mice appears to be concomitant with the mitigation of neuroinflammatory cytokine production (CD54, IL-3, CXCL9/10, and CCL2/4) [99,103] and inconsistently with or without the reduction of Tau protein staining [104,105]. Kim et al. also showed a similar anti-inflammatory effect after delivering 5 sessions of 1.8 Gy each to mice heads, associated with decreased synaptic degeneration and neuronal loss [103].

However, Ceyzériat et al. reported that memory performance improvements observed in AD mice after LDRT could also occur without changes in the amyloid load [97]. Then, it is possible that senile plaque accumulation would not be a cause but a consequence of AD [106]. Hence, to date, no molecular mechanism of LDRT’s effect on AD can be identified.

### 4.3. Degenerative Diseases and the RIANS Model

The RIANS model has already been validated in a subset of neurodegenerative diseases, such as Huntington’s disease, tuberous sclerosis, some forms of xeroderma pigmentosum D and neurofibromatosis type 1, to explain the (group II) radiosensitivity associated with these syndromes [107,108,109,110]. In all these diseases, the X-proteins (huntingtin, tuberin, XPD, and neurofibromin, respectively) are highly expressed in the cytoplasm in skin fibroblasts of the mutated patients. All these X-proteins interact with ATM, which delays the RIANS [107,108,109,110]. Interestingly, the brain is one of the organs that expresses the least ATM protein, which may suggest that the biological consequences of delayed RIANS may be more severe in the brain than in skin cells [111].

Since the 1980s, some studies have shown that fibroblasts and lymphocytes from patients with AD are radiosensitive and may have dysfunctions in some DSB repair pathways [112,113,114]. Very recently, we have identified one of the potential X-proteins associated with AD: the APOE protein is overexpressed in a large subset of AD patients and interacts with ATM by forming perinuclear ATM–pAPOE complexes (Berthel et al., submitted). These complexes prevent the RIANS and progressively cumulate unrecognized DSB, which leads to senescence, death, and accelerated aging (Berthel et al., submitted). Interestingly, exposure to radiation contributes to the dissociation of the ATM dimers and the ATM–pAPOE complexes, permitting thereafter some ATM monomers to recognize DNA breaks. However, the determination of the dose to optimize LDRT is in progress (Berthel et al., submitted).

The reports of Cuttler et al. raise another question: Cuttler et al. proposed LDRT involving doses ranging from 20 to 75 mGy. This dose range concerns the hormesis phenomenon [19]. Exposing cells to 20 to 75 mGy may result in producing ATM monomers and very few additional DNA strand breaks. Considering the pro-inflammatory DSB induced by permanent genotoxic stress, it is possible that LDRT, via a hormesis-like effect, promotes the monomerization of the ATM protein, its diffusion to the nucleus, and, finally, the initiation of DSB repair. Thereby, the RIANS model may explain the anti-inflammatory effect of LDRT with doses ranging from 20 to 75 mGy (Figure 1d). Further investigations are needed to consolidate this model and the crucial choice of the total dose and the dose per session in LDRT that should be efficient against neurodegenerative diseases.

## 5. Conclusions

LDRT is currently applied in oncology, in inflammation-related pathologies, and in AD, which illustrates well the discontinuity of the dose-response curve and the counterintuitive observation that a low dose may produce a biological effect higher than a certain higher dose. Even if further investigations are needed to document and optimize LDRT, the apparent paradox of some radiobiological effects specific to low doses may be explained by the same mechanistic model based on the radiation-induced nucleoshuttling of the ATM kinase.

## 6. Patents

The works and the data of the U1296 Inserm Unit regarding Alzheimer’s disease have been recently protected by a patent deposed in January 2023.

## Figures and Tables

**Figure 1 cancers-15-01482-f001:**
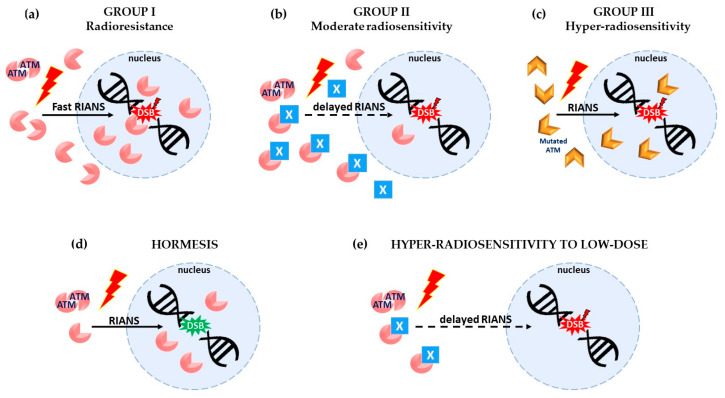
High- and low-dose radiation cell response explained by the RIANS model. The RIANS underpins the efficiency of DSB recognition and repair and the cellular radiosensitivity. At high doses, (**a**) a fast RIANS ensures radioresistance (*group I*) with complete recognition and repair of radiation-induced DSB (red star), (**b**) a RIANS delayed by the interaction of ATM with X-proteins (blue squares) leads to moderate radiosensitivity (*group II*), and (**c**) mutated ATM kinase leads to the lack of DSB recognition, which causes hyper-radiosensitivity (*group III*). It is noteworthy that group III hyper-radiosensitivity can also be caused by gross DSB repair, such as in the case of *LIG4* mutation. However, this particular case is not illustrated here. (**d**) Hormesis effect. At very low-dose (< 25 mGy), some ATM monomers can contribute to the recognition and repair of spontaneous DSB (green star), the amount of radiation-induced DSB being negligible. The hormesis effect is preferentially observed in group I cells. (**e**) Hypersensitivity to low dose (HRS). At low doses (ideally between 0.1 and 0.5 Gy), the flux of ATM monomers can be reduced again by the interactions with X-proteins (in group II cells). A too-low amount of ATM monomers that diffuse in the nucleus does not permit the recognition of the few DSB induced by radiation (red): the HRS phenomenon is preferentially observed in group II cells. It is noteworthy that the hyper-radiosensitivity observed at high doses in group III cells should not be confused with the hyper-radiosensitivity to low-dose (HRS) phenomenon. Lastly, the AR phenomenon has not been illustrated since its corresponding irradiation scenario does not fall directly into the scope of this review.

**Table 1 cancers-15-01482-t001:** State of the art of low-dose radiation therapy (LDRT) in oncology.

	Target	Study Design	Irradiation Scheme	Response	Reference
Haemato-oncology					
68 patients with NHL (90% stage III and IV)	Whole body	Retrospective	LD-TBI: midline 0.1 Gy per session (1.78 Gy total)	Recurrence-free survival: 32% at 5 years, 27% at 10 years	[30]
26 patients with NHL (stage III and IV)	Whole body	Prospective, non-randomized (pilot study)	Chemotherapy and 2 courses of LD-TBI: 5 × 0.15 Gy per session (separated by 2 weeks), followed 1 month later by radical involved-field RT (20 × 2 Gy per session)	Complete remission in 92.3% (24/26) of patients after LD-TBI and before IF-RT; Complete remission in 96.2% (25/26) of patients after IF-RT	[31]
** *Distant metastases* **					
8 patients with metastatic tumor nodules	Metastases	Randomized study	Standard fractionated RT: 12 × 1.5 Gy per session or ultra-fractionated RT: 0.5 Gy/4 h over 12 days	Significantly increased growth delay in nodules treated with the ultra-fractionated RT scheme	[32]
18 patients with metastatic colorectal cancer	Metastases	Phase II	0.2 Gy per session every 6 h interval (on each chemotherapy cycle)	Clinical or pathological complete response in 38.9% (7/18) of patients	[33]
A 73-yr woman with metastatic vaginal mucosal melanoma progressing on immunotherapy	Metastases	Case report	LDRT: 5 × 1 Gy per session (liver metastasis); 6 × 1 Gy per session (inguinal lymph node)	Clinical and radiographic complete response	[34]
74 patients with metastatic cancer (NSCLC, *n* = 38; melanoma, *n* = 21) progressing on immunotherapy within 6 months	Metastases	Phase II	HDRT alone: 3–12.5 Gy per session (20–70 Gy total) or HDRT + LDRT: 0.5–2 Gy per session (1–10 Gy total)	4-month disease control response: 47% in HDRT + LDRT vs. 37% in HDRT alone (*p* = 0.38); Overall response: 26% in HDRT + LDRT vs. 13% in HDRT (*p* = 0.27)	[35]

NSCLC = small-cell lung cancer. NHL = non-Hodgkin’s lymphoma. TBI = total body irradiation. RT = radiation therapy. HD = high dose. LD = low dose.

**Table 2 cancers-15-01482-t002:** State of the art of low-dose radiation therapy (LDRT) in inflammation-related pathologies.

	Target	Study Design	Irradiation Scheme	Response	Reference
Rheumatology					
166 patients with painful skeletal disorders (calcaneodynia, *n* = 51; achillodynia, *n* = 8; gonarthrosis, *n* = 80; bursitis trochanterica, *n* = 27)	Joints/bone	Prospective	0.5–1 Gy per session (6 Gy total)	Good response in 37.3% (62/166) of patients immediately on completion of RT and in 49.5% (54/109) of patients after a median follow-up of 29 months (*p* = 0.001)	[55]
196 patients with ankle/foot osteoarthritis	Joints	Prospective	0.5–1 Gy per session (3–6 Gy total) over 3 weeks	Subjective improvement of 80–100% in 37% (71/196) of patients	[56]
56 patients with knee/hand osteoarthritis	Joints	Randomised, sham-controlled	LDRT: 6 × 1 Gy per session or sham	No significant evidence of beneficial LDRT effect	[57]
55 patients with knee osteoarthritis	Joints	Randomised, double-blinded, sham-controlled	LDRT: 6 × 1 Gy per session or sham	No significant evidence of beneficial LDRT effect	[58]
56 patients with hand osteoarthritis	Joints	Randomised, blinded, sham-controlled	LDRT: 6 × 1 Gy per session or sham	No significant evidence of beneficial LDRT effect	[59]
**COVID-19**					
36 COVID-19 patients	Bilateral whole lungs	Prospective	1 × 0.5 Gy per session	SAFI improved from 255 mmHg to 283 mmHg at 24 h and to 381 mmHg at 1 week, respectively	[65]
41 COVID-19 patients	Bilateral whole lungs	Prospective phase I-II	1 × 1 Gy per session	SAFI significantly improved on day +3 and +7 (*p* < 0.01)	[66]
25 COVID-19 patients	Bilateral whole lungs	Phase II	1 × 0.5 Gy per session	SAFI significantly improved between pre-RT and day +2 (*p* < 0.05), +3 (*p* < 0.001) and +7 (*p* < 0.001) post-RT; oxygen supply significantly decreased between pre-RT and day +2 (*p* < 0.05), +3 (*p* < 0.001), and +7 (*p* < 0.001) post-RT	[67]
30 COVID-19 patients	Bilateral whole lungs	Multicenter, prospective, observational	1 × 0.5 Gy per session	SAFI significantly improved; oxygen supply decreased	[68]
20 COVID-19 patients	Bilateral whole lungs	Randomized, double-blinded	1 × 1 Gy per session or sham	No significant evidence in 15-day ventilator-free days (*p* = 1.00) nor overall survival at 28 days (*p* = 0.69) in both arms; lymphocyte counts significantly decreased after LDRT (*p* < 0.01)	[69]
100 COVID-19 patients	Bilateral whole lungs	Phase II, randomized	LDRT: 1 × 0.35 Gy per session or 1 × 1 Gy per session or sham	Recruiting since 2020	NCT04466683 (Ohio State University Comprehensive Cancer Center, Columbus, OH, USA)
52 COVID-19 patients	Bilateral whole lungs	Phase III, randomized	1 × 1.5 Gy per session or sham	Recruiting since 2020	NCT04433949 (Emory University Atlanta, GA, USA) [70]

RT = radiation therapy. COVID-19 = coronavirus disease 2019. SAFI = pulse oximetric saturation (SatO_2_)/fraction of inspired oxygen (FiO_2_).

**Table 3 cancers-15-01482-t003:** State of the art of low-dose radiation therapy (LDRT) in Alzheimer’s disease (AD).

	Target	Study Design	Irradiation Scheme	Response	Reference
An 81-yr-old woman with AD	Brain	Case report	5 × 40 mGy/CT over 3 months	Clinical cognitive improvement allowing discharge from hospice care	[93]
A 73-yr-old man with AD	Brain	Case report	6 × 45–50 mGy/CT over 18 months	Elevation of MMSE score from 22/30 up to 26/30	[95]
4 AD patients	Brain	Single-arm (pilot study)	4 × 40 mGy/CT over 1 month	Slight cognition and behavior improvements on quantitative measures (SIB, ADL)	[102]
30 AD patients	Brain	Phase I (single-arm pilot study)	5 × 2 Gy per session 10 × 2 Gy per session	Suspended due to staffing and budget limitations	NCT02359864 (William Beaumont Hospitals, Royal Oak, MI, USA)
5 AD patients	Brain	Phase IIa (single-arm pilot study)	5 × 2 Gy per session 10 × 2 Gy per session	Interrupted due to COVID-19	NCT02769000 (Virgina Commonwealth University, Richmond, VA, USA)
20 AD patients	Brain	Randomized, monocentric, prospective (pilot study)	5 × 2 Gy per session	Recruiting since 2017	NCT03352258 (Geneva University Hospital, Geneva, Switzerland)

MMSE = Mini-Mental State Examination. SIB = severe impairment battery. ADL = basic activities of daily living.

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
