# Peer review of "Low-Dose Radiation Therapy (LDRT) against Cancer and Inflammatory or Degenerative Diseases: Three Parallel Stories with a Common Molecular Mechanism Involving the Nucleoshuttling of the ATM Protein?"

_cancers, 2023, doi:10.3390/cancers15051482_

Round 1

Reviewer 1 Report

This paper is focused on reviewing the possibility that the so-called radiation‐induced nucleo‐shuttling of ATM protein (RIANS), a molecular mechanism studied in detail for several years by the same authors and their research team in Lyon, can describe the main features of the effects after low dose radiation therapy (LDRT). In particular, the manuscript considers three areas of application of LDRT: cancer therapy, Inflammation-related diseases, and Alzheimer disease. In effect, most of the manuscript is devoted to the review of clinical applications of LDRT.

The mechanism proposed is an interesting one, especially for its potential use in clinical radiotherapy. However, it would be desirable a more explicit mention of the possible way how to use the proposed model for improving radiotherapy.  

Also, some revision is suggested as follows.

Line 63. The words “identified and documented” should be replaced by “reviewed”, or similar, since the author cited in ref.13 does not support the occurrence of hormesis for endpoints relevant to radiation protection and related controversy about the shape of the response.

Line 68. Add “typically” before “ranging”

Lines 68-69. Please correct the challenging dose range, which is typically of the order of some Gy.

Lines 82-83. Please add ref. for the linear dose-dependence of oxidative stress.

Lines 95-96. Make it clear if group III cells are characterized by sequestration of ATM by X-proteins or by mutated ATM, Line 290.

Lines 99-100. That NO DSB are induced at doses lower than 25 mGy is not correct, since DSB formation can be regarded as a stochastic process, at least as far as dose dependence is concerned, due to the stochastic nature of energy deposition (for low-let radiation). In effect, DSB have been detected at doses as low as few mGy (e.g., Rothkamm and Loebrich PNAS 2003).

Lines 111-113. The cited paper (ref 20) does not rule out AR in group I cells. Why group II exhibits AR and also HRS (that are apparently opposite effects) should be explained.

Line 114. “Hypersensitivity”. Here group II cells are mentioned to manifest hypersensitivity while elsewhere is said that group III cells manifest HRS, as illustrated in Fig. 1

Line 137, Fig.1 can be more informative and consistent with the text if the mechanism for adaptive response were also illustrated.

 Line 221. (> Gy/min): please specify.

Line 225. HRS-positive: make it clear the link with cell grouping (I, II, III).

Line 245. As the story of radium dial painters lasted several decades I suggest not using the word "episode".

Line 264. Here relevant reference(s) and (approx.) date should be specified. For example, WHO counts about 6.8 million deaths as of Feb 3, 2023.

Line 265. Ref 58 is not very relevant in this context, while others cited by it, such as Huang et al Lancet 2020, can be appropriate.

Lines 344-45. “a tens” ?.

Lines 468. Again, as in lines 99-100: DNA strand break production is not an "all-or-nothing" process in this context and in the mentioned dose range it cannot be excluded that some, albeit few, breaks are produced.

References. Check if the format complies with the Journal rules since they recommend avoiding the abbreviated style: XXX YYY et al.

Author Response

Reply to the reviewer 1

We thank the reviewer for his/her comments.

This paper is focused on reviewing the possibility that the so-called radiation‐induced nucleo‐shuttling of ATM protein (RIANS), a molecular mechanism studied in detail for several years by the same authors and their research team in Lyon, can describe the main features of the effects after low dose radiation therapy (LDRT). In particular, the manuscript considers three areas of application of LDRT: cancer therapy, Inflammation-related diseases, and Alzheimer disease. In effect, most of the manuscript is devoted to the review of clinical applications of LDRT.

The mechanism proposed is an interesting one, especially for its potential use in clinical radiotherapy. However, it would be desirable a more explicit mention of the possible way how to use the proposed model for improving radiotherapy.  

OK. We agree. With regard to RT in general, the RIANS model has permitted to predict radiosensitivity of healthy tissues reliably, to provide biological interpretation of the LQ model and  the relationship between RBE and LET. However, these features are not in the scope of the manuscript proposed here and we have deliberately chosen to focus on LDRT (some more general references are however cited like Berthel et al. Cancers 2019).

Also, some revision is suggested as follows.

Line 63. The words “identified and documented” should be replaced by “reviewed”, or similar, since the author cited in ref.13 does not support the occurrence of hormesis for endpoints relevant to radiation protection and related controversy about the shape of the response.

See modified text.

Line 68. Add “typically” before “ranging”

See modified text.

Lines 68-69. Please correct the challenging dose range, which is typically of the order of some Gy.

  1. You are right. We apologize for this error. See modified text.

Lines 82-83. Please add ref. for the linear dose-dependence of oxidative stress.

See new references.

Lines 95-96. Make it clear if group III cells are characterized by sequestration of ATM by X-proteins or by mutated ATM, Line 290.

See modified text.

Lines 99-100. That NO DSB are induced at doses lower than 25 mGy is not correct, since DSB formation can be regarded as a stochastic process, at least as far as dose dependence is concerned, due to the stochastic nature of energy deposition (for low-let radiation). In effect, DSB have been detected at doses as low as few mGy (e.g., Rothkamm and Loebrich PNAS 2003).

We agree. See modified text.

Lines 111-113. The cited paper (ref 20) does not rule out AR in group I cells.

See modified text.

Why group II exhibits AR and also HRS (that are apparently opposite effects) should be explained.

The AR and HRS phenomena are not opposite since after a single priming dose in the frame of AR scenario or after a single HRS dose, the number of monomers is low since the dose is low but overall is reduced because of the sequestration of ATM monomers by X-proteins. See modified text.

Line 114. “Hypersensitivity”. Here group II cells are mentioned to manifest hypersensitivity while elsewhere is said that group III cells manifest HRS, as illustrated in Fig. 1

No. In fact, there is a confusion between the hyper-sensitivity of group III that is observed at high dose (at 2 Gy for example) and the phenomenon called “hypersensitivity to low dose and abbreviated HRS that is observed in a low-dose range. See modified text in the caption of Fig1.

Line 137, Fig.1 can be more informative and consistent with the text if the mechanism for adaptive response were also illustrated.

We deliberately chosen to omit the AR phenomenon in the Fig 1 to avoid any confusion since the AR irradiation scenario is not related to LDRT. See modified text in the caption of Fig1.

 Line 221. (> Gy/min): please specify.

See modified text.

Line 225. HRS-positive: make it clear the link with cell grouping (I, II, III).

  1. See modified text.

Line 245. As the story of radium dial painters lasted several decades I suggest not using the word "episode".

See modified text.

Line 264. Here relevant reference(s) and (approx.) date should be specified. For example, WHO counts about 6.8 million deaths as of Feb 3, 2023.

See modified text.

Line 265. Ref 58 is not very relevant in this context, while others cited by it, such as Huang et al Lancet 2020, can be appropriate.

See modified text.

Lines 344-45. “a tens” ?.

See modified text.

Lines 468. Again, as in lines 99-100: DNA strand break production is not an "all-or-nothing" process in this context and in the mentioned dose range it cannot be excluded that some, albeit few, breaks are produced.

See modified text.

References. Check if the format complies with the Journal rules since they recommend avoiding the abbreviated style: XXX YYY et al.

See re-formatted references

Reviewer 2 Report

In this review, the authors provide a recent update on low-dose radiation therapy (LDRT) for the treatment of cancer or non-malignant diseases such as inflammatory and degenerative diseases. In particular, the clinical application of LDRT for pulmonary complications was gaining attention during the COVID-19 outbreak. The authors propose radiation-induced nucleoshutting of the ATM kinase as a common mechanism for the effectiveness of LDRT treatment in these different diseases. This is a well written and very comprehensive review. There are some minor issues that need to be addressed before publication.

1.    On Fig.1, (b) and (e) cartoons look identical, but the explanation differs. It is confusing whether these cartoons explain hyper-sensitivity to high-dose or low-dose irradiation. Among the three phenomena described on page 3, the “Adaptive Response” scenario is missing in Fig. 1. 

2.    On page 5, line 172: the authors introduced a phase II trial by Patel et al. showing the combined effect of high-dose and low-dose RT. The patients enrolled in the study received immunotherapy, which is not mentioned in the text.

3.    On page 9, line 288: there are more biological hypotheses about the anti-inflammatory effect of LDRT on acute lung injury. Recent two publications (Jackson MR et al. Int J Radiat Oncol Biol Phys (2022) 197-211; Meziani ete al. Int J Radiat Oncol Biol Phys (2021) 1283-1294) need to be included to further explain biological mechanisms.

4.    On page 5, line 176: dis-ease is a typo.

5.    On page 5, lines 315-316: alpha and beta should be modified as Greek symbols.

6.    On page 11, table 3: “73-yo. man” is 73 yr?

7.    On page 17, line 693: “Sujin Kim, et al” is Kim S, et al. 

Author Response

Reply to the reviewer 2

We thank the reviewer for his/her comments.

In this review, the authors provide a recent update on low-dose radiation therapy (LDRT) for the treatment of cancer or non-malignant diseases such as inflammatory and degenerative diseases. In particular, the clinical application of LDRT for pulmonary complications was gaining attention during the COVID-19 outbreak. The authors propose radiation-induced nucleoshutting of the ATM kinase as a common mechanism for the effectiveness of LDRT treatment in these different diseases. This is a well written and very comprehensive review. There are some minor issues that need to be addressed before publication.

  1. On Fig.1, (b) and (e) cartoons look identical, but the explanation differs. It is confusing whether these cartoons explain hyper-sensitivity to high-dose or low-dose irradiation. Among the three phenomena described on page 3, the “Adaptive Response” scenario is missing in Fig. 1.

You are right. See modified figure and caption. With regard to AR, we deliberately chosen to omit the AR phenomenon in the Fig 1 to avoid any confusion since the AR irradiation scenario is not related to LDRT. See modified text in the caption of Fig1. We mentioned AR in the text since it is not dissociable from the hormesis and HRS phenomena and is currently cited in the phenomena that are specific to low-dose.

  1. On page 5, line 172: the authors introduced a phase II trial by Patel et al. showing the combined effect of high-dose and low-dose RT. The patients enrolled in the study received immunotherapy, which is not mentioned in the text.

See modified text.

3. On page 9, line 288: there are more biological hypotheses about the anti-inflammatory effect of LDRT on acute lung injury. Recent two publications (Jackson MR et al. Int J Radiat Oncol Biol Phys (2022) 197-211; Meziani ete al. Int J Radiat Oncol Biol Phys (2021) 1283-1294) need to be included to further explain biological mechanisms.

See modified text and new references.

  1. On page 5, line 176: dis-ease is a typo.

OK see modified text.

  1. On page 5, lines 315-316: alpha and beta should be modified as Greek symbols.

OK see modified text.

  1. On page 11, table 3: “73-yo. man” is 73 yr?

OK see modified text.

  1. On page 17, line 693: “Sujin Kim, et al” is Kim S, et al. 

OK see modified text.

Reviewer 3 Report

A concise review mainly focusing on the role of shuttling a master regulator of DNA double-strand break-induced damage response, ATM, in hormesis, adaptive response (AR) and hyper-radiosensitivity (HRS) is presented. Even the so called RIANS concept was mainly or even uniquely delved by the group of the senior author of this manuscript, an in big parts balanced review is provided which includes several new thoughts and challenges. As one example, the interconnection between radiation-induced cytokine release and their putative phosphorylation by ATM is worked-out very comprehensively.

However, for readers being no experts in radiation biology several terms and concepts should be outline in more detail.

Specifically:

-          Explain in more detail the linear quadratic model

-          Discuss in more detail concepts of liner dose responses versus non-linear ones in the low dose range

-          Explain SQ/TQ domains and the role of X-proteins in more detail

The heading of the review should be more specific and ATM shuttling should be e.g. included.

In the abstract it is misleading that in oncology in former times the number of sessions was reduced. That would mean in our times “hypofractionated” RT with a higher single dose/(remaining)fraction.

Again, in the abstract for non-radiobiologists the rationale for focusing on ATM has to be mentioned/explained more clearly.

In the paragraph about clinical data about LDRT against cancer it should be stressed more clearly that the available data are mostly based on case reports and trials with low patient numbers.

In the paragraph about LDRT and Covid some critical work on this should at least shortly be discussed and some overview work cited, as e.g. Rödel et al, PMID: 32388805.

In the mechanistic explanation of LDRT-induced amelioration of inflammation, osteoimmunological aspects with focus on interaction of immune system and bone metabolism should at least shortly be discussed with recent literature.

Regarding the very comprehensive and novel part about LDRT and Alzheimer Disease it is a bit confusing even though best effects of disease improvement was observed after exposure to a dose in the mGy range that clinical trials are then reviewed with a single dose per fraction of 2Gy.

In the sum, the topic of this important and timely review is of great relevance for readers of cancers, but several radiobiological aspects have to be clarified more

Author Response

Reply to the reviewer 3

We thank the reviewer for his/her comments.

A concise review mainly focusing on the role of shuttling a master regulator of DNA double-strand break-induced damage response, ATM, in hormesis, adaptive response (AR) and hyper-radiosensitivity (HRS) is presented. Even the so called RIANS concept was mainly or even uniquely delved by the group of the senior author of this manuscript, an in big parts balanced review is provided which includes several new thoughts and challenges. As one example, the interconnection between radiation-induced cytokine release and their putative phosphorylation by ATM is worked-out very comprehensively.

However, for readers being no experts in radiation biology several terms and concepts should be outline in more detail.

Specifically:

-          Explain in more detail the linear quadratic model

OK see modified from line 216. However, we did not wish entering in the mathematical details of the LQ formula.

-          Discuss in more detail concepts of liner dose responses versus non-linear ones in the low dose range

OK see modified text from line 64.

-          Explain SQ/TQ domains and the role of X-proteins in more detail

OK see modified text from line 98.

The heading of the review should be more specific and ATM shuttling should be e.g. included.

OK see modified title.

In the abstract it is misleading that in oncology in former times the number of sessions was reduced. That would mean in our times “hypofractionated” RT with a higher single dose/(remaining)fraction.

You are right. See modified text in abstract.

Again, in the abstract for non-radiobiologists the rationale for focusing on ATM has to be mentioned/explained more clearly.

You are right. See modified text in abstract.

In the paragraph about clinical data about LDRT against cancer it should be stressed more clearly that the available data are mostly based on case reports and trials with low patient numbers.

You are right See modified text from line 201.

In the paragraph about LDRT and Covid some critical work on this should at least shortly be discussed and some overview work cited, as e.g. Rödel et al, PMID: 32388805.

You are right See modified text from line 318.

In the mechanistic explanation of LDRT-induced amelioration of inflammation, osteoimmunological aspects with focus on interaction of immune system and bone metabolism should at least shortly be discussed with recent literature.

You are right See modified text from line 285.

Regarding the very comprehensive and novel part about LDRT and Alzheimer Disease it is a bit confusing even though best effects of disease improvement was observed after exposure to a dose in the mGy range that clinical trials are then reviewed with a single dose per fraction of 2Gy.

  1. You are right See modified text from line 456

In the sum, the topic of this important and timely review is of great relevance for readers of cancers, but several radiobiological aspects have to be clarified more

See the other corrections required by the other reviewers and the yellow-highlighted modified text.